# Equine Incisor Lesions: Histologic Confirmation of Radiographic, Macroscopic, and Micro-Computed Tomographic Findings

**DOI:** 10.3390/vetsci9070348

**Published:** 2022-07-11

**Authors:** Louisa Albers, Astrid Bienert-Zeit, Carsten Staszyk

**Affiliations:** 1Clinic for Horses, University of Veterinary Medicine Hannover, Foundation, 30559 Hannover, Germany; astrid.bienert@tiho-hannover.de; 2Faculty of Veterinary Medicine, Institute of Veterinary-Anatomy, -Histology and -Embryology, Justus-Liebig-University Gießen, 35392 Gießen, Germany; carsten.staszyk@vetmed.uni-giessen.de

**Keywords:** incisors, EOTRH, tooth resorption, histology, radiography, µCT

## Abstract

**Simple Summary:**

Incisor diseases are a common problem in equine medicine. However, Equine Odontoclastic Tooth Resorption and Hypercementosis (EOTRH) and other lesions are often only diagnosed in advanced stages. In this study, the incisors of 20 horses were examined. The findings of X-rays, macroscopical inspection, and micro-computed tomography (µCT) were compared. Five categories from healthy to severely affected teeth were formed and selected teeth from each category were additionally examined by means of histology. Odontoclastic resorption of dental hard substances, infiltration of inflammatory cells, areas of irregular dental cementum, and formation of granulation tissue were observed. Resorptive lesions previously detected by µCT were confirmed by microscopical imaging, however, not all resorptive lesions were regarded as a pathological condition like EOTRH. Detailed microscopical inspection revealed the presence of so-called surface resorptions which represent repaired resorptive lesions without any pathological consequence for the horse. Nevertheless, some incisors which appeared healthy on the X-rays featured histological findings related to EOTRH. Therefore, the detection of early incisor lesions in equine medicine remains challenging.

**Abstract:**

Equine Odontoclastic Tooth Resorption and Hypercementosis (EOTRH) and other incisor lesions are often diagnosed only in advanced stages. The incisors of 20 horses were examined radiographically, macroscopically, and via micro-computed tomography (µCT) to discriminate EORTH-affected teeth. Five categories from healthy to severely affected teeth were formed and teeth from each category were examined histologically to evaluate the opportunity of earlier radiographic diagnosis. Histologically, odontoclastic resorptive lesions, leukocytic infiltrations, and areas of irregular cementum and granulation tissue were observed. The extent and severity of histological findings were correlated to the µCT data. Micro-CT imaging was suitable to detect subtle irregularities in the dental substances which were referred to as resorptive lesions. Although histological examinations confirmed the presence of resorptive lesions, not all of them were classified as pathological conditions. Instead, repaired surface lesions were documented which were regarded as a physiological condition. Nevertheless, incisors which were radiographically regarded as healthy can also feature histological signs of EOTRH. Therefore, due to the possibility of misinterpreting radiographic findings combined with superimpositions on intraoral radiographs, the detection of early resorptive lesions remains challenging.

## 1. Introduction

The health of equine cheek teeth is of highest importance with regard to normal food comminution and nutrition [1,2]. Therefore, it is not surprising that numerous studies in the field of equine dentistry are focused on the diagnosis and therapy of equine cheek teeth diseases and wear abnormalities [3,4]. However, recognition of pathological changes in the equine incisor arcades is widely limited to traumatic injuries, developmental abnormalities (including brachygnathia), and Equine Odontoclastic Tooth Resorption and Hypercementosis (EOTRH) [3,5,6,7]. EOTRH is a progressive and often painful disease which causes several destructive processes in the teeth and their periodontal surrounding [8,9]. Hitherto, palliative therapies, such as tooth brushing and oral irrigation, dietary modifications, floating and incisor reduction, anti-inflammatory and/or antibiotic drugs, corticosteroids, surgical curettage, and debridement [3,9,10], show limited to no success in stopping the disease progression [10,11]. Thus, extraction of the affected teeth, oftentimes the complete incisor dentition, is recommended [12,13].

In histopathological examinations, inflammatory cells and inflamed tissues are present [8,14], therefore it is assumed that anti-inflammatory treatments can be beneficial in early stages of EOTRH. Consequently, it is of critical importance to identify reliable signs of early EOTRH and discriminate non-EOTRH related pathologies.

For the evaluation of incisor disorders, a clinical examination is mostly combined with radiography. Although EOTRH can have a distinct radiographic appearance [8,15], early recognition of subtle lesions is challenging [7,9,16].

Our first study [7] evaluated the reliability of radiographic imaging for diagnosing early incisor lesions. The radiographic findings were correlated with clinical and macroscopic evaluations and the teeth were also scanned and analyzed by micro-computed tomography (µCT). In addition teeth that showed pathological changes in all three modalities, numerous teeth appeared healthy in the radiographic and macroscopic evaluation but showed lesions in the µCT scans. Therefore, we suggested to validate those µCT findings by histologic examinations. The question raised whether all lesions, which were addressed as resorptive lesions on µCT scans, were correlated with histopathological changes, in terms of EOTRH. Histological features of EOTRH include resorptive lesions featuring Howship lacunae with active odontoclasts, necrotic debris, and granulation tissue as well as deposition of irregular cementum with wavy basophilic incremental lines [8,14].

Similar types of external resorptive lesions can be found in brachydont teeth. External inflammatory root resorption or external cervical root resorption are pathologic conditions which are not self-limiting and therefore indicate specific treatment [16,17,18]. These disorders share pathological characteristics with EOTRH in terms of odontoclastic driven resorptive lesions extending through the cementum into the dentin. Further, the periodontal surrounding shows infiltration of inflammatory cells and formation of granulation tissue [19].

A transient resorptive lesion in brachydont teeth is the external surface resorption. This phenomenon is not considered pathologically [17,19,20]. This type of lesion occurs on the dental roots and is a self-limiting process of resorption followed by a repair mechanism [16,19]. Histologically, areas of external surface resorption appear as small, superficial resorptive lacunae, extending into the cementum and outer layer of dentin and become subsequently repaired with newly formed cementum [19]. For this type of resorptive lesion, no treatment is indicated according to recommendations in human dental medicine [16,19].

Although resorptive lesions are most commonly diagnosed radiographically in clinical human dental practice [20], the different types of resorption need to be confirmed histologically [18].

We hypothesize that self-limiting external surface resorptions can be found in equine incisors on the palatal and lingual side of the root and might be erroneously diagnosed as pathologic or EOTRH-related lesions, especially in µCT images which display structural changes in high resolutions.

The aim of the study was to discriminate EOTRH lesions from non-EOTRH related structural changes in suspicious incisors by histological evaluation and to compare corresponding µCT and radiographic findings.

## 2. Materials and Methods

The incisors of 20 horses of different breeds (9 warmbloods, 3 Icelandic horses, 3 draft horses, 2 Haflingers, 1 Arabian horse, 1 Friesian horse, and 1 Welsh pony) and different sexes (15 mares, 4 geldings, 1 stallion) were examined. The horses were euthanized in the Clinic for Horses of the University of Veterinary Medicine Hannover or slaughtered for reasons unrelated to this study.

All horses were grouped into three age groups according to their ID documentation: (I) 9–14 years, (II) 15–19 years, and (III) 20 years and older.

After a clinical inspection of the incisor region, intraoral dorsoventral radiographs were taken (GIERTH HF 1000 (Gierth X-Ray international GmbH, Riesa, Germany); FUJIFILM DR-ID 300CL APL Software V11.0 (FUJIFILM Europe GmbH, Düsseldorf, Germany)) and the incisors (*n* = 236) were evaluated using a scoring system with the parameters describing tooth shape, structure and surface. For details, please see Albers et al. [7]. Each single tooth was scored individually by four examiners, the mean of their scorings was calculated and each tooth was sorted into one of the following groups: healthy, suspicious, moderate EOTRH, or severe EOTRH. The examiners were blinded to the horses’ signalment.

All incisors were then extracted with routine dental extraction instruments and macroscopically evaluated. For the macroscopic evaluation a scoring system with the three parameters describing tooth shape, surface, and structure was applied. For details, please see Albers et al. [7]. One of the authors (A.B.-Z.), who was blinded to the horses’ identity, evaluated the extracted incisors. Each tooth was once again sorted into one of the following groups: healthy, suspicious, moderate EOTRH, or severe EOTRH.

Subsequently, the extracted teeth were scanned in a micro-computed tomography system (QuantumFX, PerkinElmer, Waltham, MA, USA) using an isotropic voxel size of 80 µm. The obtained µCT data sets were visualized with Scry (Scry v6.0 Kuchel & Sautter GbR, Bad Teinach-Zavelstein, Germany). Cross-sectional images were reviewed for signs of dental resorption or hypercementosis. Due to the observed lesion localizations, surface meshes representing the palatal/lingual area from 5–50% of the tooth’s length were created. For details, please see Albers et al. [7]. These meshes were processed with MeshLab (ISTI—CNR, version 2016.12, Pisa, Italy) and surface curvature was computed to evaluate the surface of the incisors. Teeth which showed extraction related lesions (*n* = 50) were excluded from further evaluations. A scoring system based on the surface curvature data sets, the distribution of the surface roughness and bulbous enlargements of the apical region was applied, and the teeth were grouped into the following categories: healthy, suspicious, moderate EOTRH, or severe EOTRH.

The radiographic, macroscopic, and µCT evaluations were compared and the teeth (*n* = 186) were grouped into five final categories (see Table 1).

A representative subset of teeth from each category (*n* = 18) was further investigated by means of stereomicroscopic and histological examinations. Those 18 teeth belonged to 13 horses of 6 different breeds (4 warmbloods, 3 draft horses, 2 Haflingers, 1 Arabian horse, 1 Welsh pony, and 1 Icelandic horse) with their age ranging from 9 to 27 years (median 18.5 years). All horses were mares.

The histological examinations were focused on regions which appeared suspicious on the cross-sectional µCT scans.

The teeth were stored in neutral buffered formalin and were sectioned using a diamond-coated water-cooled band saw (Proxxon Typ MBS 240/E No 27 172, Föhren, Germany). The surfaces were assessed using a stereo microscope (ZEISS Stemi 2000-C, Carl Zeiss Jena, Germany) and regions of abnormal hard substance stratigraphy and areas of dental resorption as well as hypercementotic areas were recorded. The specimens were then decalcified in buffered ethylene diamine tetra-acetate (EDTA, pH 8.0, room temperature) for 8–10 weeks. Afterwards, the specimens were embedded in paraffin wax, 7µm thick sections were cut and stained with Toluidine blue and Picrosirius red. The sections were examined using light microscopy (Leica DM750 and Leica DM2500, Wetzlar, Germany, ocular magnification 10×, objectives 4×, 10×, 20×, 40×) and polarized light to additionally visualize collagen fiber arrangement. Physiological histological findings featured dental cementum and dentine, separated by the dentino-cemental junction, attached remnants of the periodontal ligament and pulpal tissue. Pathological findings include odontoclastic resorptive lesions, regions of deposited irregular cementum, presence of granulation tissue, or leukocytic infiltrations. For details, please see Figure 1, Figure 2, Figure 3, Figure 4, Figure 5, Figure 6 and Figure 7.

A microscope camera (Leica ICC50 HD, Wetzlar, Germany) was used to obtain microscopic photographs.

A further Category ‘0′ defined by normal physiological findings in all diagnostic modalities including histology was added because even in Category 1, histological resorptive lesions could be observed.

## 3. Results

Table 2 shows the distribution of teeth according to the different diagnostic categories. Most of the teeth were assigned to Category 2, featuring no radiographic changes but showing mild alterations macroscopically and in the µCT evaluation. Only in Age Group I, most of the teeth were assigned to Category 1.

For detailed distribution of teeth into the different categories, please see Appendix A.

Evaluating the cross-sectional µCT images, signs of dental resorption were found in 75/186 teeth. Those resorptive lesions were almost entirely found on the palatal/lingual tooth surface (*n* = 74/75). Only one tooth showed a resorptive lesion on its labial surface. Bulbous enlargement of the dental cementum on the palatal/lingual side was observed in 32/186 teeth. The dental hard substances (enamel, dentine, and cementum) were identified due to different gray scales and stratigraphic alignment. Changes in texture, stratigraphic alignment, and gray scale within the cementum and/or dentine were addressed as suspected areas of irregular cementum.

Almost all (*n* = 15/18) investigated teeth showed histological changes in terms of odontoclastic resorptive lesions, surface resorptions, areas of irregular cementum, presence of granulation tissue, or leukocytic infiltrations. The extent and severity of the described histological features was compared to findings in µCT and visualized in Figure 2, Figure 3, Figure 4, Figure 5, Figure 6 and Figure 7. Figure 1 displays a tooth described as Category ‘0′, which has no abnormalities or pathologic lesions.

The surface analysis of the µCT datasets revealed different patterns of surface irregularities. One pattern showed a streaky distribution of infoldings and protrusions, whereas the other pattern displayed a spherical distribution of irregularities (see Figure 4 and Figure 5). Comparing the histological findings with the µCT surface analysis, 7/18 specimen showed histological features of EOTRH and all of them display spherical surface irregularities.

## 4. Discussion

This was the first study validating resorptive lesions and structural changes of dental hard substances in equine incisors seen on µCT scans. In a previous study, these µCT datasets were used to verify radiographic diagnoses of resorptive and especially EOTRH-related lesions [7]. Thus, due to large discrepancies between the radiographic and µCT diagnoses, histologic examinations served as a gold standard to describe and evaluate the type and extend of dental resorption.

Whereas incisors from severely affected horses have been evaluated histologically before [8,14], this is also the first study attempting to evaluate early resorptive lesions. This study aimed to give a first descriptive evaluation and therefore no statistical conclusions could be drawn. The histologically examined teeth were selected following their radiographic, macroscopic, and µCT-based evaluation to examine teeth from each category. The authors also selected healthy and obviously affected teeth, but most of the selected teeth showed indefinite lesions on the µCT scans or contradictory findings in the three diagnostic modalities. The selected teeth represented the wide range of the horses’ age from 9 to 27 years. Thus, it can be shown, that young teeth can also display structural changes on a histological level. All horses were mares. Therefore, gender-related differences could not be evaluated. Such differences were, however, not expected, since with regard to EOTRH, no clinical, radiographic, or histological differences between male and female horses have been reported [11,14,16]. Nevertheless, the non-blinded selection of teeth holds a risk for bias and in further studies, a large sample size with randomly selected horses and teeth could reduce this risk.

Although almost all histologically examined teeth displayed structural changes of the dental hard substances, most of those changes were not radiographically detectable. Nevertheless, the clinical relevance of those lesions remains debatable.

In human dental medicine, the clinical relevance of lesions depends on the type of resorption. Tooth resorption is defined as a state associated with a physiological or pathological stimulus which results in a loss of dental hard substances or even bone [21]. The resorption is caused by odontoclasts or osteoclast-like cells [22]. However, these cells are not able to resorb unmineralized dental substances [23]. Thus, without an initial stimulus, intact dental roots can resist dental resorption due to their external and internal protective barriers formed by predentine and precementum [20].

The predominately underlying resorptive process, e.g., inflammation and/or odontoclastic resorption, is used to categorize the different types of dental resorption. The localization of the resorptive stimulus is also used to classify dental resorption into internal resorption, where the resorptive stimulus originates within the pulp, and external resorption with a stimulus originating from the periodontal ligament [20]. Therefore, the following types of tooth resorption are differentiated: external surface resorption, external inflammatory resorption, external replacement resorption, external cervical root resorption, internal inflammatory resorption, and internal replacement resorption [19,22].

External surface resorption is considered to be the least destructive form of external root resorption, to be transient and self-limiting. It is usually not visible on radiographs and features no clinical symptoms. Histologically, superficial resorptive lacunae in the outer layers of cementum and dentine can be observed, which become repaired with newly formed cementum. [19] The main etiologic factor causing this type of resorption in humans is pressure due to orthodontic treatment, impacted teeth, tumors, or cysts [22,23,24]. Pressure forces induce necrosis within the periodontal ligament and within these necrotic areas inflammatory processes lead to recruitment and activation of clastic cells which start to resorb the dental hard substances [24]. In brachydont teeth, the localization of external surface resorptions depends on the area where the initial stimulus exerts excessive pressure on the dental root area [23]. While pressure due to orthodontic treatments is distributed evenly over a root area and remains limited to a tolerable level and stimulates a well-balanced remodeling of the alveolus which results in a slow shifting of a tooth, tumors or impacted teeth exhibit more focal areas of pressure [23].

Several equine incisors in this study (*n* = 8/15) showed histological features of external surface resorptions. In contrast to brachydont teeth, the obtained lesions were almost exclusively localized in the apical third on the palatal/lingual surface suggesting a marked topographical predisposition in equine incisors for the occurrence of resorptive processes. This observation might be explained by results from biomechanical studies which located the highest pressure forces in exactly those areas of the periodontal ligament, which were identified to show surface resorptions in this study [25]. It shall be emphasized that external surface resorptions in equine incisors were revealed by µCT but remained undetected in intraoral radiographs.

Another self-limiting type of dental resorption in brachydont teeth is the transient apical breakdown, which follows moderate injury to the periodontal ligament and/or pulp [26]. Transient apical breakdown can only be found in mature teeth with fully developed roots. Radiographically, a transient widening of the periodontal ligament space and blunting of the apex can be observed. [19]

These findings can also be observed in equine radiographs and are routinely used for the characterization of pathologic and especially EOTRH-related lesions [7,15]. However, considering the transient nature of this type of lesion in brachydont teeth, the reliability of the radiographic criteria of blunted root tips and widened periodontal space should be re-evaluated in equine medicine.

In this study, the self-limiting nature of the transient apical breakdown in aged teeth could not be evaluated due to the study design. Follow-up radiographic studies are required to investigate whether a similar type of resorption is also present in equine teeth.

External inflammatory resorption is the most common type of external resorptive lesions in brachydont teeth. The initial stimulus can be traumatic, due to orthodontic treatment or periodontal infection and triggers an inflammatory process and hampers the progress of dental repair mechanisms in terms of production of reparative cementum [19]. This initiated odontoclastic resorption can be self-limiting and result in an external surface resorption or transient apical breakdown while the pulp is still vital [22]. The odontoclastic resorption will stop after repairing the damaged root surface but if the infection extends into the pulp, bacterial toxins might migrate via the dentinal tubules and the resorption will progress [22]. Clinical symptoms resemble those of periapical periodontitis, radiographically periradicular radiolucencies can be observed and an endodontic treatment is indicated [22]. Histologically bowl-shaped resorptive areas, Howship lacunae, inflammation of the periodontal ligament with granulation tissue, and infected or necrotic pulp tissue can be observed [19].

If the pathological stimulus leads to extensive periodontal necrosis and proliferating bone is overlapping dental repair mechanisms, external replacement resorption can be observed and results in a direct attachment of the tooth to the alveolar bone, a condition which is referred to as ankylosis [19,20]. This process starts with trauma and results in necrosis of periodontal ligament cells. An odontoclastic resorption of cementum and dentine is followed by osteoblastic activity gradually replacing the dental root with bone. [19,22,27]

In this study, no incisors showed signs of replacement resorption or ankylosis and to the authors’ knowledge these types of dental resorption have never been described in equine teeth. This observation might underline the high capacities of the equine periodontium to maintain and remodel a periodontal space occupied by a periodontal ligament. These capacities are much more required in a hypsodont dentition, which features continued dental eruption, than in a brachydont dentition which is characterized by a static position within its alveolus.

External cervical root resorption is a pathological process with unknown etiology in human dental medicine and is assumed to be multifactorial [22,24,28]. It is difficult to diagnose in the asymptomatic early stages, because periodontal or pulpal symptoms mainly occur in advanced stages of the progressive disease [22]. The clinical, radiographic, and histological findings of this phenomenon resemble those of Feline Odontoclastic Resorptive Lesions (FORL) [29]. External cervical root resorption might be misdiagnosed as subgingival caries, pulpitis, internal resorption, or periodontitis [22,29]. Histologically, this phenomenon features odontoclastic resorptive cavities, fibro-vascular tissue which progresses into fibro-osseous lesions, and inflammatory cells following bacterial invasion [29]. Treatment including debridement and restoration with or without endodontic therapies is required [20,30].

Although this type of dental resorption shares pathological feature not only with FORL but also with EOTRH, the localization of lesions seems to differ between the species. As equine teeth do not exhibit a tooth cervix due to their hypsodont nature, lesions in a location similar to FORL and external cervical root resorption should be expected close to the gingival sulcus in the horse. However, in this study, EOTRH-related lesions as well as external surface resorptions were reliably found in the apical third of the teeth. The same observation has been reported in the literature [8,25].

The term internal root resorption describes a condition in brachydont teeth which is characterized by inflammation and resorption within the pulp. Internal root resorptions are relatively rare in brachydont teeth [17,19,22]. It is usually a consequence of occlusal pulp exposure due to chronic pulpal inflammation, physical or chemical trauma, or bacterial infection [22]. In most cases, the apical pulp remains vital, whereas the coronal pulp becomes necrotic. In certain cases, the pulp tissue becomes replaced by newly formed hard tissues resembling bone or cementum, which is referred to as internal replacement resorption [19,22]. Root canal treatments are indicated for teeth affected by both types of internal root resorption [20].

Although no data exist that document the occurrence and/or prevalence of internal root resorption and internal replacement resorption in the horse, these findings might become confused with external lesions on radiographs as reported for brachydont species [19,22].

Diagnosis of dental root resorption in human dental medicine as well as in small animal medicine is achieved by a combination of clinical examination and radiography and early diagnosis is crucial for the management of affected teeth [18,20,31]. Especially in cats, radiography is essential in diagnosing resorptive lesions [31].

In equine medicine, the diagnosis of incisor pathologies is also achieved with clinical and radiographic examinations but early diagnosis of resorptive lesions is challenging due to superimpositions [7]. A main pathological process of EOTRH is odontoclastic inflammatory tooth resorption [9]. While these resorptive processes resemble lesions seen in brachydont teeth [16], the equine response to this process is different to the brachydont dentition. When odontoclastic resorption stops, a following healing phase in brachydont teeth ends with either cementum or bone-producing cells covering the defect in the dental substances and therefore resolves either as an external surface resorption or an external replacement resorption and/or ankylosis [23]. Lesions in the equine dentition are also able to resolve as an external surface resorption but often equine teeth produce large amounts of irregular cementum leading to hypercementotic areas. The irregular cementum features an irregular alignment of intrinsic collagen fiber bundles, irregular incremental lines, and contains numerous vascular channels of varying sizes. In response to unknown conditions, the production of irregular cementum can turn into a production of regular cementum which incorporates newly formed periodontal collagen fibers, and thus provides re-attachment of the tooth [8].

In this study, the histologically examined teeth sorted into Category 1–3 appeared radiographically healthy, but many of them displayed histological abnormalities. One tooth even displayed areas of irregular cementum. EOTRH-related resorptive lesions as well as repaired external surface resorptions were reliably found on the palatal/lingual surface in the apical third of the tooth. Thus, subtle lesions remain radiographically undetected due to superimpositions [7]. According to the study by Schrock et al. [25], these regions correspond with areas of high stresses and strains, especially in older horses. In comparison, dental resorption of brachydont teeth, especially external surface resorption and external inflammatory resorption, is mainly caused by trauma and/or pressure induced by orthodontic treatment or periodontal infection [19,22]. Thus, the changing distribution of forces in the dentition of aging horses might resemble the artificial forces of orthodontic treatments in human dental medicine.

Therefore, it can be hypothesized that changed forces on dental surfaces and periodontal ligament induce inflammatory processes with recruitment of clastic cells resulting in dental resorption. These resorptive lesions might resolve and will appear as external surface resorption, but if additional factors interfere with its repairing mechanisms, these lesions might be a starting point for the development of EOTRH. Several contributing factors such as bacterial invasion, endocrinological disorders, excessive floating, feeding management, and genetic predisposition were proposed [32,33,34].

In human dental medicine, no treatment is recommended for external surface resorption and transient apical breakdown, while external replacement resorption and ankylosis are considered untreatable [16,19,20]. In cats and dogs, the pain and discomfort caused by tooth resorption and resulting treatment recommendations are still not fully evaluated [18,31].

EOTRH-related incisor lesions share histological features with external inflammatory resorption and external cervical root resorption. In brachydont teeth, external inflammatory resorption can be self-limiting unless the pulp is involved. The clinical symptoms and disease progression of those two resorptive processes depend on the involvement of pulp and periodontal ligament. [22] Similar findings can be observed on equine patients suffering from EOTRH. Horses with large resorptive areas and pulpal involvement seem to experience more pain than horses showing mainly hypercementotic lesions. Therefore, treatment plans can vary depending on the radiographic appearance of the EOTRH-affected incisors. Due to the histologically observed involvement of inflammatory cells, treatment with NSAIDs might be attempted. However, to the authors’ knowledge, these treatments do not show sufficient success.

In the histological validation of the observed dental surface roughness and irregularities, it is challenging to ensure that the examined areas correspond to the lesions seen on the µCT scans. Nevertheless, 8/15 teeth, featuring resorptive lesions in the histological evaluation, exhibit areas of repaired surface resorption. The other 7/15 teeth featured distinct signs of EOTRH-related pathologies. Remarkably, all of those EOTRH-affected teeth displayed areas of spherical surface roughness in the µCT analysis, whereas other, especially older, teeth without EOTRH-related lesions featured a streaky irregular dental surface. In further research, the hypothesis that EOTRH features a different pattern of resorptive lesions on the tooth surface than physiological age-related processes should be evaluated.

Thus, in vivo follow-up examinations with µCT would be beneficial, but this is currently not possible. Examinations with µCT are limited to extracted teeth, whereas clinically used CT scanners do not provide the required spatial resolution to detect subtle changes of the dental surface.

## 5. Conclusions

Equine incisors feature a marked topographical predisposition for initial resorptive lesions on the palatal/lingual side of the root, as visualized on µCT scans and in histological examinations. These lesions might resolve in clinically irrelevant external surface resorption but might progress to early phases of EOTRH.

Unfortunately, radiography appears insufficient to detect these lesions in equine incisors due to superimpositions.

## Figures and Tables

**Figure 1 vetsci-09-00348-f001:**
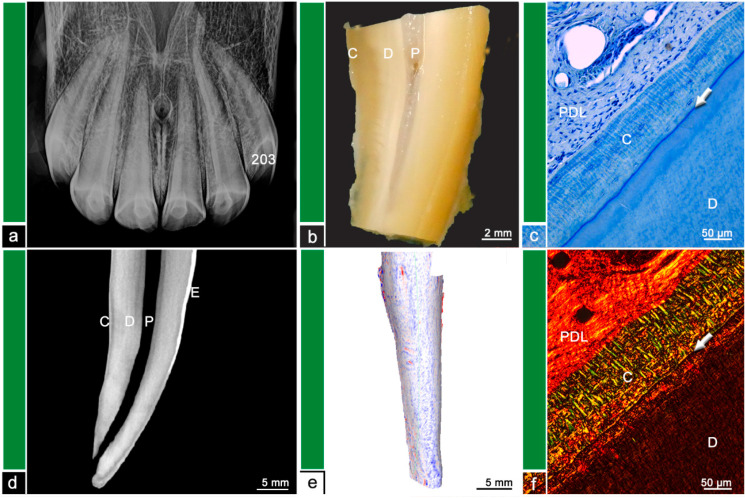
Category ‘0’, tooth 203, 12-year-old mare. (**a**) Intraoral radiograph. Tooth 203 does not show any radiographic findings. (**b**) Stereomicroscopic image. Although the exact border between cementum (C) and dentine (D) is not visible most peripheral areas of the dental hard substances are defined as cementum and areas near the pulp (P) are defined as dentine. (**c**) Histological image, toluidine blue stain. The white arrow indicates the dentino-cemental junction, which appears as a straight, continuous blue line. D = dentine, C = cementum, PDL = periodontal ligament. (**d**) Cross-sectional µCT image. Although the exact border between cementum (C) and dentine (D) is not visible most peripheral areas of the dental hard substances are defined as cementum and areas near the pulp (P) are defined as dentine. The dental surface appears regular and smooth with no irregularities. E = Enamel. (**e**) Mesh reconstruction of µCT image with visualized surface curvature. The dental surface appears regular and smooth with only minimal irregularities. White areas = smooth surface, red color = invaginations of the surface and blue color = protrusions of the surface. (**f**) Histological image, picrosirius red stain. The white arrow indicates the dentino-cemental junction. The cementum (C) features a well-organized mesh of parallel aligned collagen fiber bundles. D = dentine, PDL = periodontal ligament.

**Figure 2 vetsci-09-00348-f002:**
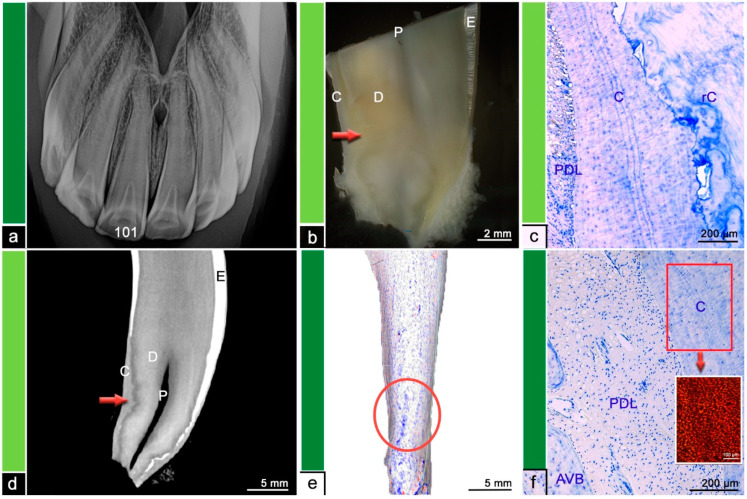
Category 1, tooth 101, 10-year-old mare. (**a**) Intraoral radiograph. Tooth 101 does not show any radiographic findings. (**b**) Stereomicroscopic image. Although the exact border between cementum (C) and dentine (D) is not visible most peripheral areas of the dental hard substances are defined as cementum and areas near the pulp (P) are defined as dentine. The red arrow indicates a suspected resorption filled with reparative cementum. (**c**) Histological image, toluidine blue stain. A mass of reparative cementum (rC) is overlayed by regular cementum (C) providing dental attachment to the periodontal ligament (PDL). (**d**) Cross-sectional µCT image. Although the exact border between cementum (C) and dentine (D) is not visible most peripheral areas of the dental hard substances are defined as cementum and areas near the pulp (P) are defined as dentine. The tooth surface appears regular and smooth. The red arrow indicates a suspected resorption filled with reparative cementum. E = Enamel. (**e**) Mesh reconstruction of µCT image with visualized surface curvature. The dental surface appears regular and smooth with only minimal irregularities. Within the red ellipse the region corresponding to the suspected resorption in (**d**) displays slightly more surface irregularities. White areas = smooth surface, red color = invaginations of the surface and blue color = protrusions of the surface. (**f**) Histological image, toluidine blue stain. C = Cementum, PDL = periodontal ligament, AVB = alveolar bone. Inset, picrosirius red stain. The displayed cementum features a well-organized mesh of parallel aligned collagen fiber bundles providing attachment to the periodontal fiber apparatus.

**Figure 3 vetsci-09-00348-f003:**
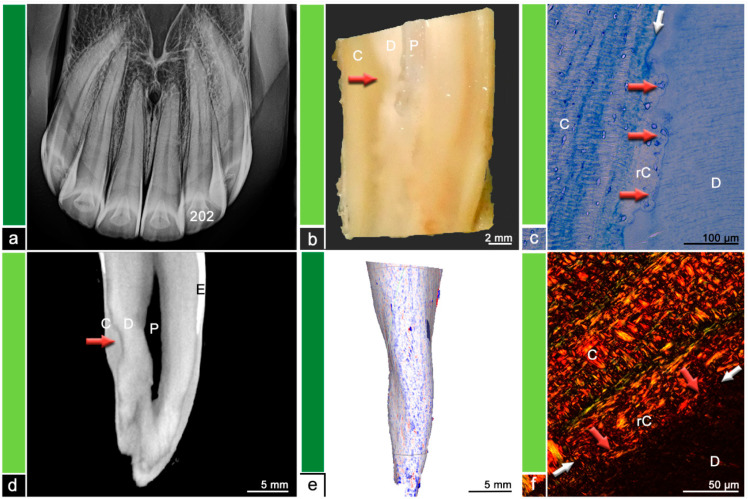
Category 2, tooth 202, 19-year-old mare. (**a**) Intraoral radiograph. Tooth 202 does not show any radiographic findings. (**b**) Stereomicroscopic image. Although the exact border between cementum (C) and dentine (D) is not visible most peripheral areas of the dental hard substances are defined as cementum and areas near the pulp (P) are defined as dentine. The red arrow indicates a suspected resorption filled with reparative cementum. (**c**) Histological image, toluidine blue stain. The red arrows indicate a resorptive lesion filled with reparative cementum (rC). The white arrow shows the regular dentino-cemental junction. D = dentine, C = cementum. (**d**) Cross-sectional µCT image. Although the exact border between cementum (C) and dentine (D) is not visible most peripheral areas of the dental hard substances are defined as cementum and areas near the pulp (P) are defined as dentine. The red arrow indicates a suspected resorption filled with reparative cementum and a slight hypercementotic enlargement. E = Enamel. (**e**) Mesh reconstruction of µCT image with visualized surface curvature. The dental surface appears regular and smooth with only minimal irregularities. White areas = smooth surface, red color = invaginations of the surface and blue color = protrusions of the surface. (**f**) Histological image, picrosirius red stain. The red arrows indicate a resorptive lesion filled with reparative cementum (rC) featuring an unorganized aligned collagen fiber arrangement compared to the collagen fiber arrangement displayed in the regular cementum (C). The white arrows show the level of the regular dentino-cemental junction. D = dentine.

**Figure 4 vetsci-09-00348-f004:**
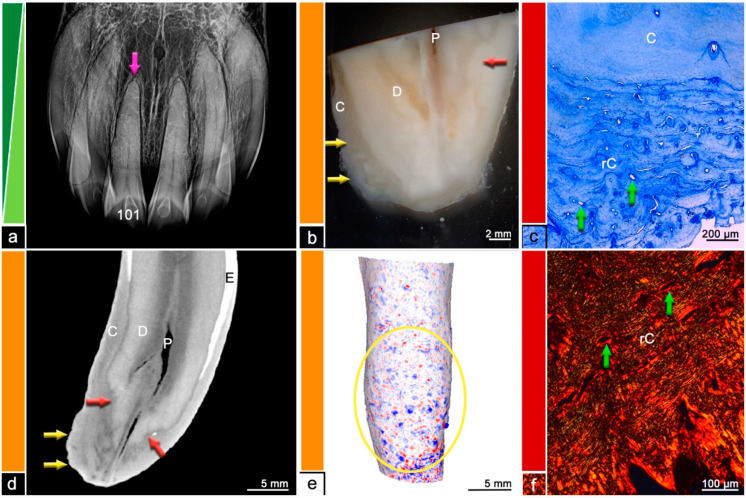
Category 3, tooth 101, 22-year-old mare. (**a**) Intraoral radiograph showing tooth 101 with a slightly blunted root tip (pink arrow). (**b**) Stereomicroscopic image. Although the exact border between cementum (C) and dentine (D) is not visible most peripheral areas of the dental hard substances are defined as cementum and areas near the pulp (P) are defined as dentine. The red arrow indicates a suspected resorption filled with reparative cementum. The yellow arrows display the enlarged root. (**c**) Histological image, toluidine blue stain. Masses of reparative cementum (rC) cause a hypercementotic enlargement. The reparative cementum features wavy incremental lines and large vascular channels (green arrows). C = cementum. (**d**) Cross-sectional µCT image. Although the exact border between cementum (C) and dentine (D) is not visible most peripheral areas of the dental hard substances are defined as cementum and areas near the pulp (P) are defined as dentine. The red arrows indicate suspected resorptions filled with reparative cementum. The yellow arrows display the bulbous enlargement of the root tip. E = Enamel. (**e**) Mesh reconstruction of µCT image with visualized surface curvature. The dental surface appears irregular and especially within the yellow ellipse spherical irregularities can be observed. White areas = smooth surface, red color = invaginations of the surface and blue color = protrusions of the surface. (**f**) Histological image, picrosirius red stain. Reparative cementum (rC) features an irregular arrangement of collagen fibers and numerous vascular channels (green arrows).

**Figure 5 vetsci-09-00348-f005:**
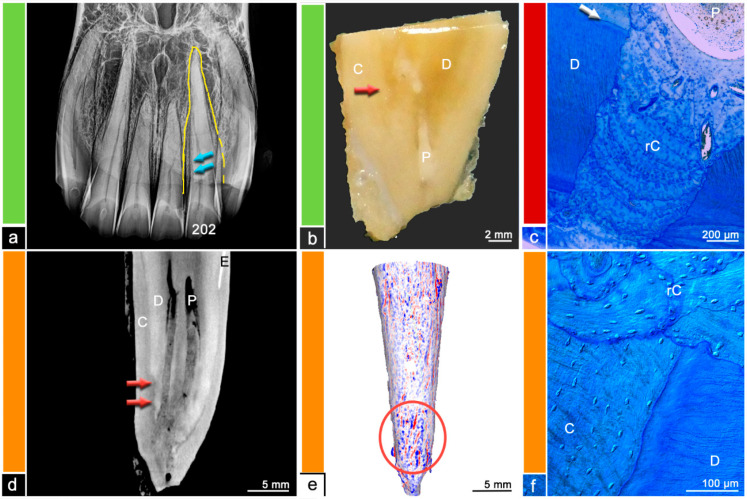
Category 4, tooth 202, 19-year-old mare. (**a**) Intraoral radiograph. The yellow line indicates an enlargement of the intra-alveolar part of the tooth. Blue arrows indicate mild surface irregularities. (**b**) Stereomicroscopic image. Although the exact border between cementum (C) and dentine (D) is not visible most peripheral areas of the dental hard substances are defined as cementum and areas near the pulp (P) are defined as dentine. The red arrow indicates a suspected resorption filled with reparative cementum. (**c**) Histological image, toluidine blue stain. Reparative cementum (rC) extends into the pulp cavity (P). The white arrow indicates the regular dentino-cemental junction. D = dentine, C = cementum. (**d**) Cross-sectional µCT image. Although the exact border between cementum (C) and dentine (D) is not visible most peripheral areas of the dental hard substances are defined as cementum and areas near the pulp (P) are defined as dentine. The red arrows indicate suspected resorptions filled with reparative cementum. E = Enamel. (**e**) Mesh reconstruction of µCT image with visualized surface curvature. The dental surface appears irregular and especially within the red ellipse a streaky distribution of irregularities can be observed. White areas = smooth surface, red color = invaginations of the surface and blue color = protrusions of the surface. (**f**) Histological image, toluidine blue stain. Masses of reparative cementum (rC) feature an irregular arrangement of incremental lines and an irregular distribution of cementoblasts. D = dentine, C = cementum.

**Figure 6 vetsci-09-00348-f006:**
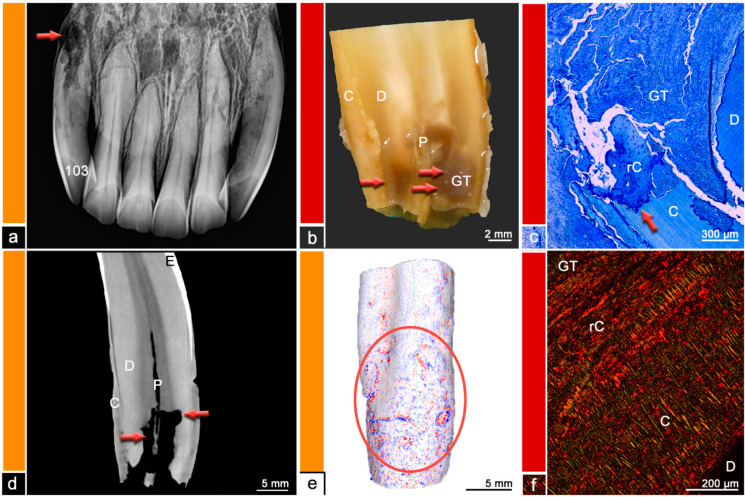
Category 5, tooth 103, 27-year-old mare, resorptive type of EOTRH. (**a**) Intraoral radiograph. The tooth 103 displays a large apical resorption (red arrow). (**b**) Stereomicroscopic image. Although the exact border between cementum (C) and dentine (D) is not visible most peripheral areas of the dental hard substances are defined as cementum and areas near the pulp (P) are defined as dentine. The red arrows indicate a suspected resorption filled with granulation tissue (GT). (**c**) Histological image, toluidine blue stain. Dentine (D), cementum (C), reparative cementum (rC) and a mass of granulation tissue (GT) can be observed. The red arrow indicates a resorptive lesion filled with reparative cementum (rC). (**d**) Cross-sectional µCT image. Although the exact border between cementum (C) and dentine (D) is not visible most peripheral areas of the dental hard substances are defined as cementum and areas near the pulp (P) are defined as dentine. The red arrows indicate resorptive areas. E = Enamel. (**e**) Mesh reconstruction of µCT image with visualized surface curvature. The dental surface appears irregular and especially within the red ellipse spherical irregularities can be observed. White areas = smooth surface, red color = invaginations of the surface and blue color = protrusions of the surface. (**f**) Histological image, picrosirius red stain. Reparative cementum (rC) with its irregular arrangement of collagen fibers compared to regular cementum (C) can be observed. D = dentine, GT = granulation tissue.

**Figure 7 vetsci-09-00348-f007:**
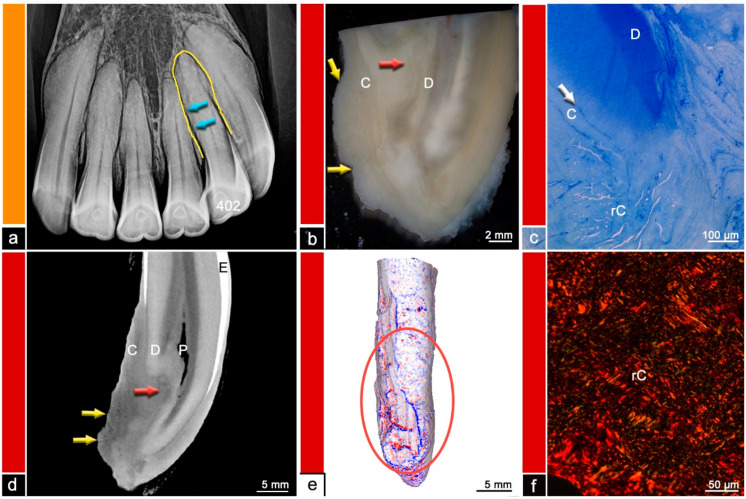
Category 5, tooth 402, 23-year-old mare, hypercementotic type of EOTRH. (**a**) Intraoral radiograph. The yellow line indicates a mildly enlarged intra-alveolar part of the tooth. The blue arrows indicate mild surface irregularities. (**b**) Stereomicroscopic image. Although the exact border between cementum (C) and dentine (D) is not visible most peripheral areas of the dental hard substances are defined as cementum and areas near the pulp are defined as dentine. The red arrow indicates a suspected resorption filled with reparative cementum. The yellow arrows display the enlarged root. (**c**) Histological image, toluidine blue stain. Large areas of reparative cementum (rC) can be observed. The white line shows the regular dentino-cemental junction. D = dentine (D), C = cementum. (**d**) Cross-sectional µCT image. Although the exact border between cementum (C) and dentine (D) is not visible most peripheral areas of the dental hard substances are defined as cementum and areas near the pulp (P) are defined as dentine. The red arrow indicates suspected resorptions filled with reparative cementum. The yellow arrows display the bulbous enlargement of the root tip. E = enamel. (**e**) Mesh reconstruction of µCT image with visualized surface curvature. The dental surface appears notably irregular and especially within the red ellipse large irregularities can be observed. White areas = smooth surface, red color = invaginations of the surface and blue color = protrusions of the surface. (**f**) Histological image, picrosirius red stain. Reparative cementum (rC) features an irregular arrangement of collagen fibers.

**Table 1 vetsci-09-00348-t001:** Categories according to the radiographic, macroscopic, and µCT diagnoses.

Category	Radiography	Macroscopy	µCT Surface Analysis
1	healthy	healthy	healthy
2	healthy	one or both suspicious, ≤one moderate	
3	healthy	both ≥ moderate, or one severe and the other healthy or suspicious	
4	suspicious or moderate	both ≠ severe	
5	≥ suspicious	and	≥one severe

**Table 2 vetsci-09-00348-t002:** Distribution of teeth in the five categories.

Category	All Teeth	Age Group (I)	Age Group (II)	Age Group (III)
1	19 (10.2%)	17 (53.1%)	0 (0.0%)	2 (2.4%)
2	85 (45.7%)	14 (43.8%)	36 (50.7%)	35 (42.2%)
3	25 (13.5%)	0 (0.0%)	16 (22.5%)	9 (10.8%)
4	27 (14.5%)	1 (3.1%)	9 (12.7%)	17 (20.5%)
5	30 (16.1%)	0 (0.0%)	10 (14.1%)	20 (24.1%)
	186 (100%)	32 (100%)	71 (100%)	83 (100%)

## Data Availability

The datasets generated for this study are available on request to the corresponding author.

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
