# Peer review of "Equine Incisor Lesions: Histologic Confirmation of Radiographic, Macroscopic, and Micro-Computed Tomographic Findings"

_vetsci, 2022, doi:10.3390/vetsci9070348_

Round 1

Reviewer 1 Report

This manuscript builds on the Albers et al 2022 EVJ paper by including histology. It is of interest to equine researchers and clinicians. Specific comments are as follows:

Line 84, “validate” connotes statistical analysis. Since none was performed, perhaps “compare” is a better work choice.

Line 131, since the aim of the study is to discriminate EORTH vs. non-EORTH teeth via histology, it would be useful if in the methods there was a definition of the histological finding in each of these categories.

Results section. It would be useful to include a table like the EVJ Table 2 that has three additional columns – clinical assessment, stereomicroscopic and histological categorization. This would be potentially more informative than the current Table 2.

Line 161 “correlated” connotes statistical analysis. Since none was performed, perhaps there is a better work choice.

Line 162 “category 0” is not defined in the methods. Please define if you plan to use.

Figures 1-7 are well done. Could you add a little bar below each figure with a color that corresponded to the severity of the lesion in each category: the clinical assessment, radiograph, macro-assessment, micro-CT, stereo histo and histo. For example, in figure 1, each bar would be green. In figure 2, the radiograph bar would be green, but the other bars would light green, orange, etc, depending on severity of the lesion as identified by that modality.

From a clinical perspective, it is very interesting that some lesions are inflammatory (contain leukocyte infiltrations) as mentioned in the abstract. Please also mention in the results section. In the discussion section, please speculate based on what is known in other species, if NSAID treatment vs. tincture of time in horses with suspicious EORTH lesions would be indicated. Perhaps something to pursue in a follow up longitudinal study?

Line  220: The different types of tooth resorption is human dentistry are defined here. The subsequent paragraphs further describe these categories and relate them to what was found histologically in this study. It would be useful if the authors could include the number of teeth in this study that fit these categories. From a clinical perspective, it would be great to know how many fit into the self-limiting types and again would be interesting to pursue in a follow up longitudinal study.

Line 388: Are there clinical assessment features that practitioners would be advised of monitoring to help determine the early signs of EOTRH even before radiographic changes?

Author Response

This manuscript builds on the Albers et al 2022 EVJ paper by including histology. It is of interest to equine researchers and clinicians. Specific comments are as follows:

Line 84, “validate” connotes statistical analysis. Since none was performed, perhaps “compare” is a better work choice.

Answer: Thank you for your recommendation. We changed it accordingly.

Line 131, since the aim of the study is to discriminate EORTH vs. non-EORTH teeth via histology, it would be useful if in the methods there was a definition of the histological finding in each of these categories.

Answer: Thank you for your comment. The categories were defined prior to histological examinations and without inclusion of the histological findings. The aim was to verify the µCT findings and thereby evaluate the opportunity of radiographic early detection. Thus, no histological findings were defined prior to the examinations.

Results section. It would be useful to include a table like the EVJ Table 2 that has three additional columns – clinical assessment, stereomicroscopic and histological categorization. This would be potentially more informative than the current Table 2.

Answer: Thank you for your recommendation. We had actually started with such a table, but then we realized that it would be too much information and thus too confusing and complex for the reader. Therefore,  we decided to present our results in this condensed form.

Line 161 “correlated” connotes statistical analysis. Since none was performed, perhaps there is a better work choice.

Answer: Changed it to “compare”.

Line 162 “category 0” is not defined in the methods. Please define if you plan to use.

Answer: We added a sentence about category ‘0’ in ll. 144-146.

Figures 1-7 are well done. Could you add a little bar below each figure with a color that corresponded to the severity of the lesion in each category: the clinical assessment, radiograph, macro-assessment, micro-CT, stereo histo and histo. For example, in figure 1, each bar would be green. In figure 2, the radiograph bar would be green, but the other bars would light green, orange, etc, depending on severity of the lesion as identified by that modality.

Answer: Thank you for your suggestion. We added a small bar on the left side of each panel in the figures.

From a clinical perspective, it is very interesting that some lesions are inflammatory (contain leukocyte infiltrations) as mentioned in the abstract. Please also mention in the results section. In the discussion section, please speculate based on what is known in other species, if NSAID treatment vs. tincture of time in horses with suspicious EORTH lesions would be indicated. Perhaps something to pursue in a follow up longitudinal study?

Answer: Our histological findings are summarized in ll. 161-166 and a phrase regarding treatments with NSAIDs was also added in ll. 368-370.

Line  220: The different types of tooth resorption is human dentistry are defined here. The subsequent paragraphs further describe these categories and relate them to what was found histologically in this study. It would be useful if the authors could include the number of teeth in this study that fit these categories. From a clinical perspective, it would be great to know how many fit into the self-limiting types and again would be interesting to pursue in a follow up longitudinal study.

Answer: Thank you for your advice. We added the number of teeth featuring external surface resorptions in l. 236. We examined 18 teeth, three of them were physiological, eight featured external surface resorptions and seven showed EOTRH related lesions. However, the given numbers are not suitable to derive any epidemiological conclusion. This would indeed require further studies.

Line 388: Are there clinical assessment features that practitioners would be advised of monitoring to help determine the early signs of EOTRH even before radiographic changes?

Answer: Unfortunately, to our knowledge there are no clinical findings that could reveal early EOTRH related lesions before radiography.

Reviewer 2 Report

The paper takes up the very interesting issue of EOTRH. A similar disorder also occurs in humans, dogs and particularly frequently in cats. The aetiology is not clear. Although the clinical and radiological picture is similar, the histological picture is different in different species.  The research material is sufficient. The methodology is adequate. The photographs are of very high quality and fully correspond to the text. In the discussion (in paragraph 320) that in cats, intraoral X-ray is the most important diagnostic method.

Author Response

The paper takes up the very interesting issue of EOTRH. A similar disorder also occurs in humans, dogs and particularly frequently in cats. The aetiology is not clear. Although the clinical and radiological picture is similar, the histological picture is different in different species.  The research material is sufficient. The methodology is adequate. The photographs are of very high quality and fully correspond to the text. In the discussion (in paragraph 320) that in cats, intraoral X-ray is the most important diagnostic method.

Answer: Thank you very much for your revision. We added a sentence about radiography in cats in ll. 318-319.

Round 2

Reviewer 1 Report

Thank you for addressing the majority of the previously recommended revisions. 

Line 131 Although histological changes were not defined a priori, it would be useful to define the histology that connotes a normal (category dark green), vs. abnormal (light green, orange and red categories). The authors state in the discussion, "histological examinations served as the gold standard." It would be helpful to define this gold standard in the methods. 

Line 144 add . . ."normal" physiological . . .

As noted previously, an expanded table 2 was recommended. If the editors disagree with this comment, including the following as a supplemental table would be useful for readers with interest: Albers et al EVJ 2022 Table 2 that has three additional columns – clinical assessment, stereomicroscopic and histological categorization.

Author Response

Thank you for addressing the majority of the previously recommended revisions. 

Line 131 Although histological changes were not defined a priori, it would be useful to define the histology that connotes a normal (category dark green), vs. abnormal (light green, orange and red categories). The authors state in the discussion, "histological examinations served as the gold standard." It would be helpful to define this gold standard in the methods. 

Answer: Thank you for your advice. We aimed to show the histological features in the figures rather than describing them in the text, but a paragraph about the histological findings was added in ll. 141-145.

Line 144 add . . ."normal" physiological . . .

Answer: We changed it accordingly.

As noted previously, an expanded table 2 was recommended. If the editors disagree with this comment, including the following as a supplemental table would be useful for readers with interest: Albers et al EVJ 2022 Table 2 that has three additional columns – clinical assessment, stereomicroscopic and histological categorization.

Answer: Thank you again for your recommendation. An expanded tables as you suggested is difficult to provide because the clinical score was not calculated on a single tooth basis and there are only stereomicroscopical and histological findings of 18 teeth. Nevertheless, we provided two supplementary tables which we hope are useful for the readers.